# Insights from a Bibliometric Analysis of Vividness and Its Links with Consciousness and Mental Imagery

**DOI:** 10.3390/brainsci10010041

**Published:** 2020-01-10

**Authors:** Stefanie Haustein, André Vellino, Amedeo D’Angiulli

**Affiliations:** 1School of Information Studies, University of Ottawa, Ottawa, ON K1N 6N5, Canada; stefanie.haustein@uottawa.ca; 2Centre Interuniversitaire de Recherche sur la Science et la Technologie (CIRST), Université du Québec à Montréal, Montreal, QC H3C 3P8, Canada; 3Scholarly Communications Lab., Ottawa, ON K1N 6N5, Canada; 4Department of Neuroscience, Carleton University, Ottawa, ON K1S 5B6, Canada; amedeo.dangiulli@carleton.ca; 5NICER Lab., Carleton University, Ottawa, ON K1S 5B6, Canada

**Keywords:** vividness, consciousness, mental imagery, bibliometrics, map of science, term co-occurrence

## Abstract

We performed a bibliometric analysis of the peer-reviewed literature on vividness between 1900 and 2019 indexed by the Web of Science and compared it with the same analysis of publications on consciousness and mental imagery. While we observed a similarity between the citation growth rates for publications about each of these three subjects, our analysis shows that these concepts rarely overlap (co-occur) in the literature, revealing a surprising paucity of research about these concepts taken together. A disciplinary analysis shows that the field of Psychology dominates the topic of vividness, even though the total number of publications containing that term is small and the concept occurs in several other disciplines such as Computer Science and Artificial Intelligence. The present findings suggest that without a coherent unitary framework for the use of vividness in research, important opportunities for advancing the field might be missed. In contrast, we suggest that an evidence-based framework (such as the bibliometric analytic methods as exemplified here) will help to guide research from all disciplines that are concerned with vividness and help to resolve the challenge of epistemic incommensurability amongst published research in multidisciplinary fields.

## 1. Introduction

From the inception of scientific psychology, vividness has been a key psychological construct related to the nature of the human condition and consciousness (see the pioneers such as Galton, Wundt, Hebb). Ubiquitously associated with personality and individual differences of imagination, this concept has had a dubious status and unclear definitions and ever-changing meanings continue into the present. Most recently, a prepotent revival of interest as reflected by almost 400 publications in the last five years (according to the Web of Science) in social, cognitive and neuroscientific psychology poses the challenge of understanding its uses and definitions to create theories that provide a framework for useful knowledge creation.

In many studies, mental images are either treated as conscious entities by definition, or as empirical operations implicit in completing some type of task, such as, for example, the measurement of reaction time in mental rotation [1]. In the latter context, an underlying mental image is assumed, but there is no direct determination of whether it is conscious or not. As summarized by Baars [2], “we have very little firm evidence about the conscious dimension of mental imagery”. Vividness of mental images is a construct which may be crucial in achieving this missing bridge in research, as it may correspond to consciousness or aspects of consciousness of images.

There is currently a surge of interest in vividness in the cognitive neuroscience and neuroimaging literature (see [3] for a review). Based on this literature, it seems that a general implicit assumption is that vivid images are conscious, and it is possible that the least vivid images are effectively unconscious or that they become such once a threshold (e.g., the “no image” in the Vividness of Visual Imagery Questionnaire [4]) is reached. Thus, it is still unclear whether the vividness dimension may in fact be a kind of “disguised correlate of consciousness” [2] or if, instead, it might be a supramodal metacognitive dimension not necessarily associated with imagery. However, even from studies using a vividness approach in neuroscience, the conscious dimension of mental imagery is not explicitly or fully tackled head on and the potentially critical contribution of vividness as a mediating property of consciousness is lost. Therefore, as it stands, we seem to be missing pieces in a complex relationship (between consciousness and imagery) which could be captured by a potential mediator (vividness).

To address the pressing question of what we know about the links between vividness, consciousness and imagery in the research done so far, we undertook a bibliometric and terminological analysis of the concept of *vividness* in the context of research in *consciousness* and *mental imagery*, from 1900 to 2019. We performed an analysis of publications indexed in the Web of Science (WoS) containing these terms to both generate a network graph of word co-occurrences and to explore several bibliographic characteristics of these publications over time. Our aim was to create a preliminary overview of the publication landscape of vividness and its links to consciousness and imagery. The motivation was to lay the foundations for further research with the long-term objective of building a unified framework for the cohesive use of the construct of vividness. The rationale is that such a framework could help researchers at least better understand the semantic variations in the current usage of this concept, if not help to formulate a sort of inter- and multi-disciplinary lingua franca.

## 2. Materials and Methods

The most elementary way to measure of the relevance of an idea or concept in the literature is to calculate the frequency with which peer-reviewed publications mentioning that concept are published. Queries on such terms in the WoS publication metadata of most of the extant literature provided us with these basic publications statistics. The WoS Core Collection used for our analysis includes the Science Citation Index, Social Science Citation Index and the Arts & Humanities Citation Index, which cover 21,177 peer-reviewed journals. At the time of writing, WoS included 74.9 million documents published between 1900 and 2019. 

Basic bibliometric indicators, such as the number of publications and co-occurrences, were used to assess the growth of the relevant concepts over time. The analyzed metadata is based on the bibliographic information provided in WoS, which we downloaded from the online platform in September 2019. 

Based on the bibliographic information, we constructed concept maps based on term co-occurrences in the title of these papers. We used VOSviewer, a social network analysis software tool specifically designed to extract meaningful network information from bibliographic and bibliometric metadata [5,6]. VOSviewer visualizes network data in the form of network graphs to create effective and efficient visualizations of complex relationship data. For example, one can create co-authorship networks, citation networks and co-occurrence networks. In order to restrict the text to the most meaningful terms, we used VOSviewer to extract noun phrases by applying a linguistic filter based on a part-of-speech tagger [6]. Extracted terms were analyzed and cleaned using the thesaurus function of VOSviewer. We combined acronyms with the occurrences of their long forms (e.g., VVIQ, EMDR, VMIQ, PTSD and their corresponding long form expressions), merged synonyms and quasi-synonyms such as “image vividness” and “imagery vividness” and removed extraneous adjectives in some multi-word terms (e.g., “enhanced vividness”, “high vividness”) to one significant word (e.g., “vividness”). The final network, after merging synonyms, contained 1065 individual noun phrases. Clustering and network layout in VOSviewer are based on the association strength algorithm, which normalizes co-occurrences of two items by their overall occurrences [6]. Clustering identifies the most frequently co-occurring and thus most similar terms and visualizes this similarity by assigning nodes belonging to the same cluster the same color on the map. Terms that appeared frequently in the title of the same documents, and thus have many connections to other terms, are positioned in the center of the network. Those terms that do not co-appear often with others are located at the periphery. The size of the nodes in the network graph represents the number of occurrences of a term and the thickness of edges demonstrates the number of co-occurrences between two nodes.

## 3. Results

### 3.1. Analysis of Title Terms

The numbers of documents published in the last 120 years containing the keywords of interest in the title (TI = (“vividness” or “liveliness”)/TI = “mental imag*”/TI = “consciousness”) are shown in Figure 1. Overall, 22,909 documents contained consciousness in the title were published, followed by 1381 for mental imag* (which includes “mental image”, “mental images”, “mental imaging”, etc.) and 393 documents on “vividness or liveliness”. We can assume that if mentioned in the title, that the articles focus on the respective concepts. When the concepts appear in the abstract or keyword field, they might still be relevant but can be considered less central to the article overall. 

The first publications mentioning “vividness” in the title were both published in the *Journal of Experimental Psychology* in 1929 [7] and 1932 [8]. With the exception of five other early works in in the 1940s and 1950s, the concept only starts to become relevant in the late 1960s. With some declines in 1985 and 1995, the number of publications increased overall, particularly in the mid-2000s. The number of annual publications focusing on vividness peaked in 2017 at 30 documents.

Several features of this figure deserve to be highlighted. The publication rates for each of the search terms from about 1966 onward show similar growth trends. A particular increase can be observed from the mid-2000s onwards. The increase is similar to growth rates in these same fields (Psychology/Neurosciences) for publications using cognate terms such as “mind” and “mental” (data not shown). We note, however, that there is a time lag between the times at which these concepts reached a critical threshold of relative publication rates. The terms “vividness/liveliness” only reached 10 or more publications per year (roughly 30% of the peak of 30 publications per year in 2017 after 2008) whereas “mental imag*” reached that 30% from its peak per year in 2002 and “consciousness” reached that threshold in 1975.

Looking at the overlap between documents that mention more than one of the concepts in their titles (Figure 2), it becomes apparent that none of the documents address all three concepts together. The size of each circle is proportional to the total number of documents and the size of the intersections is proportional to their co-occurrence in the literature. There are also no documents that mention both vividness (or liveliness) and consciousness in the title. Moreover, the overlap between vividness and mental imag* is low: 32 documents address both concepts in their titles, this corresponds to 8.1% of 393 vividness or 2.3% of mental imaging publications. The overlap between consciousness and mental imag* is even lower at nine documents.

On the level of the disciplinary distribution of publications, almost half (48%) of the 393 articles with “vividness” in the title were published in Psychology journals, followed by Neuroscience and Neurology (6%), Psychiatry (5%), Business and Economics (4%) and Literature (3%). On the journal level, articles were much more evenly distributed. As shown in Figure 3, the most common publication venues were the *British Journal of Psychology* (10 documents; 2.5%), *Journal of Behavior Therapy and Experimental Psychiatry* (9; 2.3%), *International Journal of Clinical and Experimental Hypnosis* (7; 1.8%) and *Memory* (6; 1.5%).

The co-occurrence network in Figure 4 shows that in the whole network of relationships there seem to be many connections between the main vividness node and the main nodes for concepts related to imagery, memory and consciousness. However, these connections are scattered and preponderantly weak. The most frequent noun phrases (as displayed by node size) were *vividness*, *liveliness*, *effect*, *imagery*, *imagery vividness*, *mental imagery*, *relationship*, *VVIQ*, *visual imagery*, *memory*, *individual difference*, *emotionality*, *eye movement*, *memory vividness* and *vividness effect*. The 1065 title terms were grouped into 67 clusters according to their co-occurrence similarity. The largest cluster (red) grouped the terms *memory* (10 document titles), *emotionality* (9), *eye movement* (9) and *memory vividness* (9). It was followed by cluster 2 (green), which contained the terms *vividness* (200), *animated image* (3), *divine power* (3), *naturalism* (3) and *roman theory* (3). Cluster 3 (blue) contained the terms *function* (6), *implication* (6) and *measure* (5). The title terms *individual difference* (10), *moderating role* (3) and *multisensory imagery vividness* (3) were the most frequent noun phrases in cluster 4 (yellow). Cluster 5 contained *impact* (5), *study* (5), *comparison* (3), *fMRI* (3) and *visual imagery vividness* (3), while *liveliness* (44) together with *privacy* (2) was assigned to cluster 6 (turquoise).

### 3.2. Analysis of Title and Abstract Terms

Expanding the title search to titles, abstracts and keywords (WoS topic search (TS field tag)) includes additional documents which refer to the concepts of vividness, mental imagery and consciousness, but these concepts might not be central to the respective publications. The expansion of the queries increases the number of relevant publications to 2186 for “vividness or liveliness”, 5203 for “mental imag*” and 76,920 to “consciousness”. We note here that there is proportionally more work published on *vividness* and *mental imagery* than there is on either *consciousness* and *mental imagery* or *consciousness* and *vividness*.

Taken together over time, the number of publications in the intersection of the search terms that co-occur in the title, abstract or keyword fields is shown in the Venn diagram in Figure 5. As little as 15 documents (0.7% of vividness publications) mention all three concepts in the title, abstract or keyword fields. These documents are listed in Table 1 in descending order of citations received. These documents were published in 14 journals between 1999 and 2019 with seven of them published since 2016, which suggests that the combination of the three concepts is gaining traction in recent years. However, the scattering of articles across 14 journals suggests that there is no “natural habitat” for these kinds of publications.

We note also that a subset of these papers argue that the link between these concepts has been neglected over the years. In [3] for example, one of the co-authors of this article argued that a systematic account of mental imagery that integrates its cognitive, affective, neural and phenomenological aspects, including vividness and consciousness is still lacking.

A tree-map (Figure 6) of the top 15 most frequent disciplines, in which journal articles containing “vividness or liveliness” were published as a topic, shows that the reach of this concept far exceeds the disciplinary boundaries of Psychology. Articles containing “vividness” have been exported to fields such as Computer Science, Business, Engineering and Education. The treemap, where the size of each rectangle represents the number of articles, is based on WoS categories, which are assigned at the journal level.

## 4. Discussion

From our results, it seems evident that a mapping of terms, concepts and constructs linked with vividness through bibliometric analysis provides valuable insights. As a researcher in the fields of Psychology or Neuroscience would expect, vividness appears linked with the subfields of imagery, memory, and clinical practice. However, it also appears that the set vividness is related to many different scattered aspects related to the set imagery/memory, and consciousness. This is evident from Figure 4. Furthermore, confirming the very motivation of our analysis, there are actually few *direct* explicit intersections between vividness, imagery/memory and consciousness, as reflected by the low number of journal articles containing more than one of the three terms (Figure 3). The historical trends of publications in Figure 1 suggest that there might be similar patterns of growth in publication, which could be explained by the common repressive influence of behaviorism on the three constructs as they represent three main examples of “mentalism” [23]. This may be evidence for why the links between these variables might have been expressed only indirectly. Whatever the cause, an implication of the scarcity of intersections between the three constructs in current research is suggestive of the challenge in integrating knowledge about them in an effective manner in the current scientific paradigm.

These findings could have a few possible interpretations. However, we argue that a plausible reason that there are many scattered links between vividness, consciousness and imagery, which are yet not as direct as one might expect, rests in the status of vividness itself as a construct. It is possible to interpret the pattern of results as showing that although vividness is at the center of so much research, it indeed is the least established concept, lagging far behind consciousness and imagery as central concepts in Psychology and Neuroscience. A plausible and parsimonious explanation for this is the polysemy of the term “vividness”. Besides the ordinary sense of “vivid” associated with clarity, intensity of hue and chromatic purity, this term is also used as synonym for relevance or salience, emotional expressivity or intensity of emotional content, strength of memory, richness of imagination or detail or meaning, and finally the ease with which memories or mental images are recalled.

The major complicating factors seem to be the surprising variety of meanings of the term vividness and how it is used or theorized. Some authors do not mention imagery or consciousness at all when using the term vividness but associate it with various forms of memory such as prospective, episodic and autobiographical memory, or to aliased processes not literally called imagery (e.g., imaginings, visualizations, simulations). Similarly, replacement constructs for vividness have been offered, for example, in terms of strength of imagery or semantic long-term memory contents, such as the general memory dimensions mentioned earlier or more specifically sensory, autobiographical or episodic memories. In other cases, vividness is replaced by synonyms such as “liveliness” (which we captured in our analysis) or the even vaguer “richness” and used in a way that is purely narrative and disjoined from previous scientific literature.

In future bibliometric research of this kind, it may be useful to expand the above analyses to include more articles without the targeted keywords. Our initial analysis made a reasonable attempt at identifying synonyms for the targeted keywords, but it was not intended to be exhaustive (for instance, some previous work used “clarity” instead of “vividness” or “liveliness”). The analyses could also go beyond the keywords in title and abstract, for instance, by including established experimental paradigms and measures for these three concepts which fall in the purview of the proposed background framework of reference.

Consequently, current research practice related to vividness resembles a Tower of Babel where researchers from different traditions talk over each other without sharing a common language, understanding or knowledge. One possible reason is that while the construct is very salient, its use might have far exceeded the disciplinary boundaries of Psychology and Neuroscience. For instance, it has been exported in fields such as Computer Science, AI and consciousness research, while simultaneously having a rich historical background of usage in many languages and non-scientific cultures (for example in poetry and literature) [24].

If the reasoning above is plausible, there is much to be gained (by researchers in the field) to consolidate the construct of vividness by looking at the meaning of the different occurrences in the various relationships that one can find in the overlapping literatures. Our analysis could be a starting point for this long-term objective. For example, it may be possible to refine the mapping created in Figure 5 to extract a core and periphery of what a normative meaning of vividness could be for multidisciplinary research in consciousness and imagery.

The product of this analysis could be a framework that could be empirically tested and therefore help researchers stir research into productive, useful and plausible scientific directions (i.e., avoiding seemingly contradictory situations of many relational associations which remain like isolated islands and do not reach full integration in a coherent wider network of theoretical statements, models and constructs). Case in point, it seems that if things remain as they are, statements about vividness cannot be easily compared empirically (i.e., they are epistemically incommensurable [25]) from one paper to the next. Yet, contrary to what one may infer from this fragmentation of knowledge, our analysis provides evidence that the construct of vividness does have some explicative power or efficacy in terms of unifying theory.

Thus, it is possible that unless we try to achieve a coherent unitary framework for the use of vividness in research, important opportunities for advancing the field might be missed. On the contrary, an evidence-based framework will help to resolve semantic ambiguities and guide research from all disciplines that are concerned with vividness.

The need for such a framework is exemplified by the limitations of the current bibliometric analysis itself. Our searches for term-occurrence in the WoS does not, in fact, guarantee a maximum amount of “recall” for the topical aboutness of publications. In other words, it is possible to write about a subject (“vividness”) without ever mentioning the exact term. Thus, in our study three peer reviewed journals, *Journal of Mental Imagery*, *Imagination Cognition and Personality* and *Memory and Cognition*, are not on the list or are not the main contributors in the list of most relevant journals but do in fact contain articles that are related to vividness even if the exact word does not appear in the metadata of the articles. It is clear that to overcome this limitation in future bibliometric analyses it is essential to have a solid conceptual framework of reference to make valid semantic attributions and inferences.

We conclude that bibliometric analysis, as demonstrated by this preliminary work and further strengthened by semantically enhanced search, could be an invaluable tool for showing future research the best way ahead.

## Figures and Tables

**Figure 1 brainsci-10-00041-f001:**
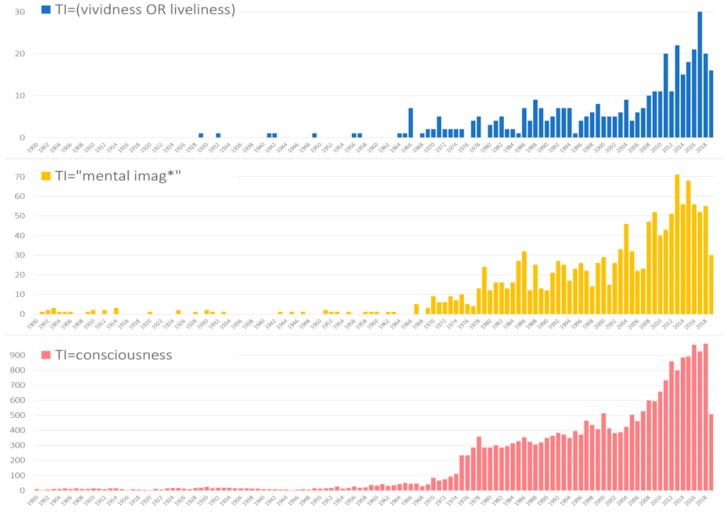
Annual publication rates in Web of Science (WoS) for documents with search terms in the title.

**Figure 2 brainsci-10-00041-f002:**
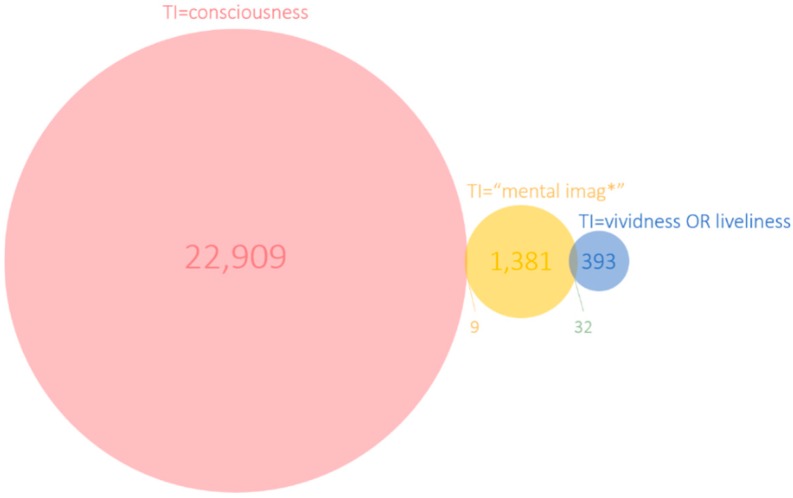
Venn diagram of the co-occurrence of terms in the title field (TI).

**Figure 3 brainsci-10-00041-f003:**
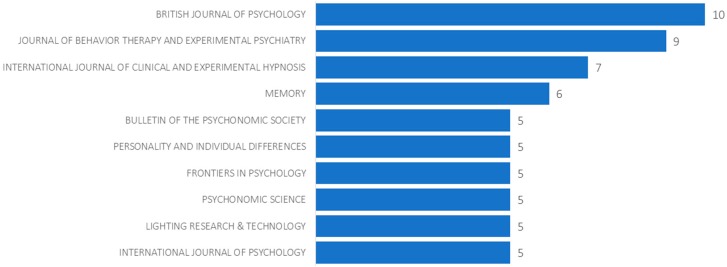
Top 10 journals publishing documents with “vividness” in the title.

**Figure 4 brainsci-10-00041-f004:**
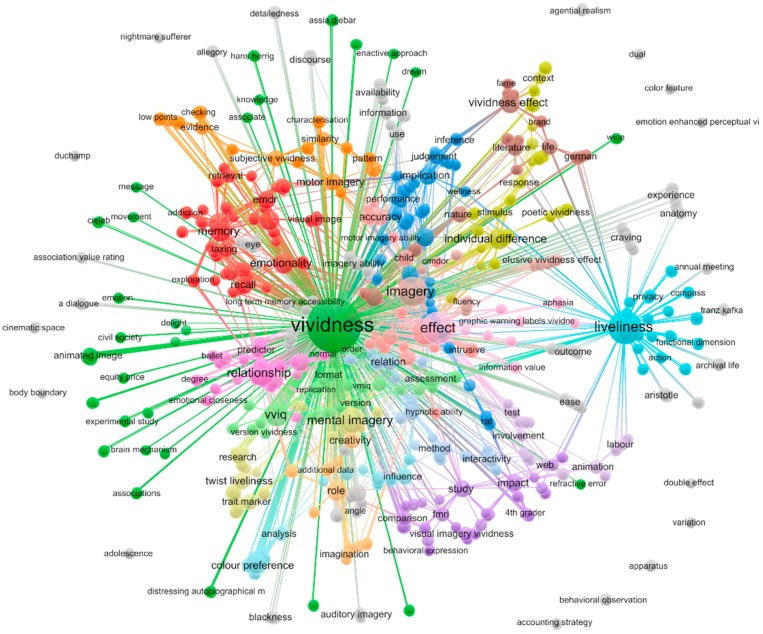
Co-occurrence network of noun phrases based on articles with “vividness” in the title.

**Figure 5 brainsci-10-00041-f005:**
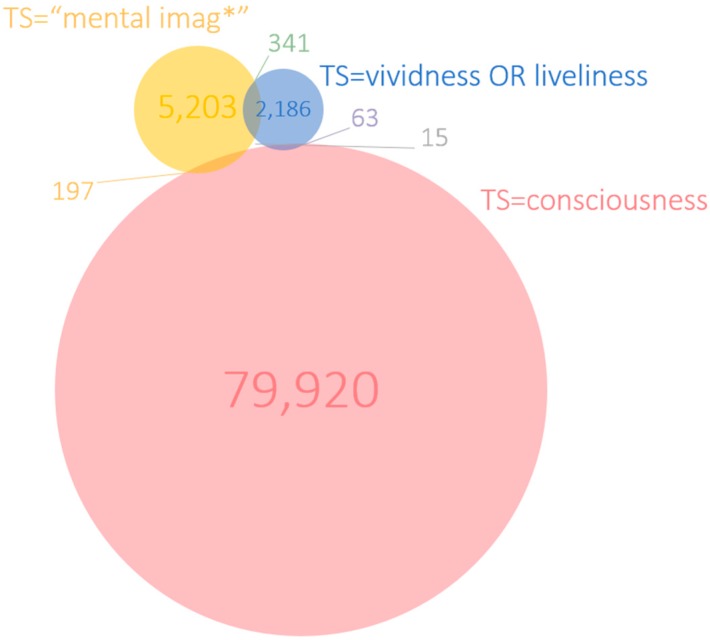
Venn diagram of the co-occurrence of terms in the title, abstract or keyword (TS) fields.

**Figure 6 brainsci-10-00041-f006:**
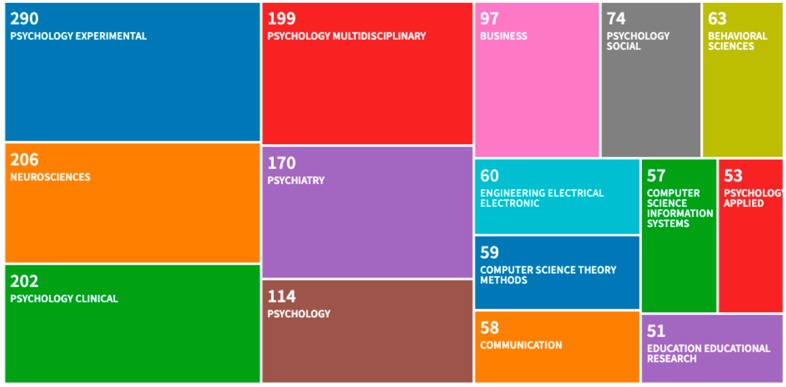
Top 15 disciplines of publications containing the term “vividness” in the title, abstract or keyword fields.

**Table 1 brainsci-10-00041-t001:** Fifteen documents’ terms in title, abstract or keywords ordered by number of citations. Matches to search terms in the title are in bold and underscored.

First Author	Title; DOI	Journal	Ref.	WoS Citations
Piolino, P.	Re-experiencing old memories via hippocampus: a PET study of autobiographical memory; 10.1016/j.neuroimage.2004.02.025	*Neuroimage*	[9]	108
Pearson, J.	Evaluating the Mind’s Eye: The Metacognition of Visual Imagery; 10.1177/0956797611417134	*Psychological Science*	[10]	59
Marks, D.F.	**Consciousness**, **mental imagery** and action; 10.1348/000712699161639	*British Journal of Psychology*	[11]	59
Lilley, S.A.	Visuospatial working memory interference with recollections of trauma; 10.1348/014466508X398943	*British Journal of Clinical Psychology*	[12]	55
Rademaker, R.L.	Training visual imagery: Improvements of metacognition, but not imagery strength; 10.3389/fpsyg.2012.00224	*Frontiers in Psychology*	[13]	28
Vianna, E.P.M.	Does vivid emotional imagery depend on body signals?; 10.1016/j.ijpsycho.2008.01.013	*International Journal of Psychophysiology*	[14]	8
Huang, M.P.	Vivid visualization in the experience of phobia in virtual environments: Preliminary results; 10.1089/10949310050078742	*Cyberpsychology & Behavior*	[15]	8
Iachini, T.	The experience of virtual reality: are individual differences in **mental imagery** associated with sense of presence?; 10.1007/s10339-018-0897-y	*Cognitive Processing*	[16]	4
Deroy, O.	Lessons of synaesthesia for **consciousness**: Learning from the exception, rather than the general; 10.1016/j.neuropsychologia.2015.08.005	*Neuropsychologia*	[17]	4
Runge, M.S.	Meta-analytic comparison of trial-versus questionnaire-based **vividness** reportability across behavioral, cognitive and neural measurements of imagery; 10.1093/nc/nix006 10.1093/nc/nix006	*Neuroscience of Consciousness*	[3]	3
Santarpia, A.	Evaluating the **vividness** of **mental imagery** in different French samples; 10.1016/j.prps.2007.11.001	*Pratiques Psychologique*	[18]	3
Fazekas, P.	White dreams are made of colours: What studying contentless dreams can teach about the neural basis of dreaming and conscious experiences; 10.1016/j.smrv.2018.10.005	*Sleep Medicine Reviews*	[19]	1
van Schie, C.C.	When I relive a positive me: Vivid autobiographical memories facilitate autonoetic brain activation and enhance mood; 10.1002/hbm.24742	*Human Brain Mapping*	[20]	0
Marks, D.E.	I Am Conscious, Therefore, I Am: Imagery, Affect, Action, and a General Theory of Behavior; 10.3390/brainsci9050107	*Brain Sciences*	[21]	0
Ribeiro, N.	Investigating on the Methodology Effect When Evaluating Lucid Dream; 10.3389/fpsyg.2016.01306	*Frontiers in Psychology*	[22]	0

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
