# Peer review of "Insights from a Bibliometric Analysis of Vividness and Its Links with Consciousness and Mental Imagery"

_brainsci, 2020, doi:10.3390/brainsci10010041_

Round 1

Reviewer 1 Report

the article is characterized by clear style, rigorous method and clarity of the scientific interpretation regarding results;

is a welcome article for researchers in the field of mental imagery;

VOSviewer is a relatively new soft for analyzing bibliometric networks and perhaps it should be an extra phrase on how to use it;

lines 138-140 -two publications containing articles related to vividness even if the word do not appear in the titles of the articles: Memory and cognition;

Journal of mental imagery; 

to highlight the value of the study, the article might answer to the question if this kind of research was done for vividness and mental imag*;

I think a comma is still needed, even in the first sentence.

Author Response

Thank you for your helpful comments.  Here are our responses.

1) VOSviewer is a relatively new soft for analyzing bibliometric networks and perhaps it should be an extra phrase on how to use it;

On lines 84-90 we have added supplementary sentences that explain what VOSviewer is and what it does as well as a reference to the user-manual for how to use this tool.

2) lines 138-140 -two publications containing articles related to vividness even if the word do not appear in the titles of the articles: Memory and cognition; Journal of mental imagery

We interpreted this comment to mean there are, in fact, two journals “Memory and Cognition” and "Journal of mental imagery” that are not on our list but do in fact contain articles that are related to vividness “even if the word does not appear in the titles of the articles”;    We agree with the reviewer and addressed this point with a new paragraph in the discussion section and now on lines 293-302, which reads:   The need for such a framework is exemplified by the limitations of the current bibliometric analysis itself. Our searches for term-occurrence in the WoS does not, in fact, guarantee a maximum amount of “recall” for the topical aboutness of publications. In other words, it is possible to write about a subject (“vividness”) without ever mentioning the exact term. Thus, in our study three peer reviewed journals, Journal of Mental Imagery, Imagination Cognition and Personality and Memory and Cognition, are not on the list or are not the main contributors in the list of most relevant journals but do in fact contain articles that are related to vividness even if the exact word does not appear in the metadata of the articles. It is clear that to overcome this limitation in future bibliometric analyses it is essential to have a solid conceptual framework of reference to make valid semantic attributions and inferences.

3) to highlight the value of the study, the article might answer to the question if this kind of research was done for vividness and mental imag*;

We also agree with this comment by Reviewer 1.    We believe that this comment is now addressed by the added paragraph currently on lines 253-259, which reads:   In future bibliometric research of this kind, it may be useful to expand the above analyses to include articles without the targeted keywords. Our initial analysis made a reasonable attempt at identifying synonyms for the targeted keywords, but it was not intended to be exhaustive (for instance, some previous work used “clarity” instead of “vividness” or “liveliness”). The analyses could also go beyond the keywords in title and abstract, for instance, by including established experimental paradigms and measures for these three concepts which fall in the purview of the proposed background framework of reference. 

Reviewer 2 Report

The manuscript assessed the potential relationship between three concepts (vividness, imagery, and consciousness) by performing bibliometric analyses of published literature on vividness and its overlapping with conscious and imagery. The analyses on the co-occurrence of these three concepts in published literature and the co-occurrence network revealed a surprising finding that there is little overlapping of the three concepts in empirical research, indicating a potentially new research direction. It addressed a timely issue regarding the roles of vividness in research on consciousness. Given the emerging interests on this topic, the manuscript would provide a unique contribution to the field with these bibliometric analyses.

It may be useful to expand the analyses to include articles without the targeted keywords. The manuscript did a reasonable job with identifying synonyms for the targeted keywords, although it  was not exhaustive. For instance, some previous work used clarity instead of “vividness or liveliness”). The analyses could also go beyond the keywords in title and abstract. For instance, to  included established experimental paradigms/measures for these three concepts.

Author Response

Thank you for your helpful comments.   You say:   It may be useful to expand the analyses to include articles without the targeted keywords. The manuscript did a reasonable job with identifying synonyms for the targeted keywords, although it  was not exhaustive. For instance, some previous work used clarity instead of “vividness or liveliness”). The analyses could also go beyond the keywords in title and abstract. For instance, to  included established  paradigms/measures for these three concepts.   We believe that this comment is also addressed by the added paragraph mentioned above and currently on lines 253-259, which reads:   In future bibliometric research of this kind, it may be useful to expand the above analyses to include articles without the targeted keywords. Our initial analysis made a reasonable attempt at identifying synonyms for the targeted keywords, but it was not intended to be exhaustive (for instance, some previous work used “clarity” instead of “vividness” or “liveliness”). The analyses could also go beyond the keywords in title and abstract, for instance, by including established experimental paradigms and measures for these three concepts which fall in the purview of the proposed background framework of reference.    We have also corrected the typographical errors noted and discovered a few more while proof-reading (also changed in the attached version)
